# Hypoxia Inhibits Cell Cycle Progression and Cell Proliferation in Brain Microvascular Endothelial Cells via the miR-212-3p/MCM2 Axis

**DOI:** 10.3390/ijms24032788

**Published:** 2023-02-01

**Authors:** Qixin Shi, Shaohua Li, Qiang Lyu, Shuai Zhang, Yungang Bai, Jin Ma

**Affiliations:** Department of Aerospace Physiology, Air Force Medical University, Xi’an 710032, China

**Keywords:** hypoxia, blood–brain barrier, miR-212-3p, MCM2, cell cycle, proliferation

## Abstract

Hypoxia impairs blood–brain barrier (BBB) structure and function, causing pathophysiological changes in the context of stroke and high-altitude brain edema. Brain microvascular endothelial cells (BMECs) are major structural and functional elements of the BBB, and their exact role in hypoxia remains unknown. Here, we first deciphered the molecular events that occur in BMECs under 24 h hypoxia by whole-transcriptome sequencing assay. We found that hypoxia inhibited BMEC cell cycle progression and proliferation and downregulated minichromosome maintenance complex component 2 (*Mcm2*) expression. *Mcm2* overexpression attenuated the inhibition of cell cycle progression and proliferation caused by hypoxia. Then, we predicted the upstream miRNAs of MCM2 through TargetScan and miRanDa and selected miR-212-3p, whose expression was significantly increased under hypoxia. Moreover, the miR-212-3p inhibitor attenuated the inhibition of cell cycle progression and cell proliferation caused by hypoxia by regulating MCM2. Taken together, these results suggest that the miR-212-3p/MCM2 axis plays an important role in BMECs under hypoxia and provide a potential target for the treatment of BBB disorder-related cerebrovascular disease.

## 1. Introduction

The blood–brain barrier (BBB) is a highly selective microvascular structure between the blood and the central nervous system (CNS) and plays a crucial role in maintaining normal brain function and CNS homeostasis [1]. The BBB is formed by brain endothelial cells (ECs) interacting with mural cells, immune cells, glial cells, and neural cells in the neurovascular unit [2,3]. As a major structural and functional element of the BBB, brain microvascular endothelial cells (BMECs) have unique properties that enable them to strictly regulate the movement of ions, molecules, and cells between blood and the brain [2,4,5,6]. BMECs are held together by tight junctions, which greatly limit the paracellular flux of solutes [7]. Hypoxia severely impairs BBB structure and function, causing pathophysiological changes [8,9,10]. BBB decomposition may lead to leukocyte infiltration into the CNS and promote damage [11]. Therefore, it is essential to understand the pathological process of BBB injury. Protecting BMECs to reduce secondary injuries may be a potential way to treat CNS diseases, including stroke and high-altitude brain edema.

MicroRNAs (miRNAs) are a group of endogenous noncoding RNA molecules with approximately 22 nucleotides that can play important regulatory roles in animals and plants by targeting mRNAs for cleavage or translational repression [12]. miRNAs can potentially regulate every aspect of cellular function, including growth, proliferation, differentiation, development, metabolism, infection, immunity, cell death, organellar biogenesis, messenger signaling, DNA repair and self-renewal, among others [13,14]. Furthermore, some miRNAs, such as miR-210, miR-21-5p, miR-4463, and miR-17-5p, are sensitive to hypoxia and exert significant effects on endothelial cells [15,16,17,18]. Previous studies show that miR-212 is highly expressed in the CNS and plays an important role in tissue development and the formation and plasticity of neuronal connections [19,20,21,22,23]. On the other hand, miR-212 was reported to serve as an oncogene or tumor suppressor by influencing different targets or pathways during oncogenesis and the development and metastasis of cancer [24]. A systematic review indicated that miR-212 could be a novel potential biomarker in cancer diagnosis and prognosis [25]. Studies showed that miR-212 could regulate tight junction stabilization in the blood–brain barrier after stroke [26,27]. However, the effects of miR-212-3p on the cell cycle and proliferation in brain microvascular endothelial cells during hypoxia have not been reported.

Minichromosome maintenance complex component 2 (MCM2) is a member of the minichromosomal maintenance family of proteins which mainly regulates DNA replication and the cell cycle by participating in the formation of the replication initiation complex [28]. Generally, MCM2–7 interact with each other to form a complex with a heterohexameric structure to perform their functions [29]. MCM2 is the most researched protein of all the MCMs. Recent studies have shown that MCM2 is significantly upregulated in 30 cancers and may be a potential marker for various cancers [30,31]. A previous study demonstrated that inhibiting MCM2 could reduce cell viability, aggravate apoptosis, and increase Tau phosphorylation in an Alzheimer’s disease cell model [32]. However, the role of MCM2 in microvascular endothelial cells during hypoxia remains unclear.

In the current study, we first deciphered the molecular events of BMECs under 24 h hypoxia by whole-transcriptome sequencing assay. We found that hypoxia inhibited cell cycle progression and proliferation in BMECs and downregulated minichromosome maintenance complex component 2 (*Mcm2*) expression. MCM2 overexpression attenuated the inhibition of cell cycle progression and proliferation caused by hypoxia. Our study further indicated that miR-212-3p was the upstream miRNA of MCM2 that could inhibit cell cycle progression and proliferation in BMECs. Our study suggests that the miR-212-3p/MCM2 axis may have therapeutic potential in treating BBB disorder-related cerebrovascular disease.

## 2. Results

### 2.1. Functional Analysis of Differentially Expressed Genes (DEGs) in Hypoxia−Exposed bEnd.3 Cells

To decipher the molecular events that occur during 24 h hypoxia exposure in bEnd.3 cells, we performed a whole-transcriptome sequencing assay. A volcano map was generated to visualize the DEGs (Figure 1A), and a heatmap was generated to visualize the top upregulated and downregulated genes (Figure 1B). A total of 1576 genes were significantly upregulated, and 3261 genes were downregulated (fold change > 2, *p* < 0.05) in bEnd.3 cells exposed to hypoxia (Appendix A).

To elucidate the roles of DEGs in different biological pathways, gene ontology (GO) and Kyoto Encyclopedia of Genes and Genomes (KEGG) enrichment analyses were performed. GO analysis revealed that the DEGs were involved in the regulation of transcription, DNA templates, immune system process, and inflammatory responses in the biological processes (BP) category; in the membrane, an integral component of the membrane and plasma membrane in the cellular component (CC) category; and in metal ion binding, hydrolase activity, and G-protein coupled adenosine receptor activity in the molecular function (MF) category (Figure 1C). Furthermore, KEGG pathway enrichment analysis revealed that the DEGs were mainly involved in the cell cycle, DNA replication, ribosome biogenesis in eukaryotes, and the hypoxia-inducible factor 1 (HIF−1) signaling pathway (Figure 1D).

Next, to further investigate molecular changes, we performed gene set enrichment analysis (GSEA) using the KEGG gene set. Bar charts were generated to visualize the top 10 upregulated (Figure 1E) and downregulated pathways (Figure 1F). The results showed that the upregulated pathways included graft versus host disease, Th1 and Th2 cell differentiation, and the HIF−1 signaling pathway, and the downregulated pathways included DNA replication, ribosome biogenesis in eukaryotes, and cell cycle. In conclusion, we performed a functional analysis of these differentially expressed genes to understand the molecular changes that occur in cells under hypoxia.

### 2.2. Hypoxia Inhibits Cell Cycle Progression and Cell Proliferation in bEnd.3 Cells

GSEA revealed that the cell cycle and DNA replication pathways were significantly downregulated in bEnd.3 cells under hypoxia (Figure 2A). RNA−seq showed that the transcription of cyclin−dependent kinase 1 (*Cdk1*), *Cdk2*, *Cdk4*, *cyclin A2*, *cyclin B1*, *cyclin D1*, and *cyclin E1* was significantly decreased (Figure 2B). To investigate whether hypoxia inhibited cell cycle progression and proliferation in microvascular endothelial cells, the expression of those cell cycle−related proteins was examined by Western blotting after hypoxia exposure for 24 h, and the results showed that their expression was significantly decreased (Figure 2C). In addition, we counted the number of cells in different phases of the cell cycle by flow cytometry. We found that hypoxia increased the proportion of cells in the G0/G1 phase and decreased the proportion of cells in the S phase (*p* < 0.01) (Figure 2D).

Then, 5−ethynyl-2′−deoxyuridine (EdU) staining was used to detect microvascular endothelial cell proliferation under hypoxia. The percentage of EdU−positive cells was significantly decreased compared with that in the control (*p* < 0.01) (Figure 2E). The Cell Counting Kit−8 (CCK-8) assay was used to evaluate cell viability. Cell growth continued to increase in a time-dependent manner, and the growth of the hypoxia group was significantly inhibited, indicating that hypoxia inhibited the proliferation of microvascular endothelial cells (Figure 2F). Proliferating cell nuclear antigen (PCNA) is a protein that acts as a processivity factor for DNA polymerase δ in eukaryotic cells [33]. Osteopontin (OPN) is a secreted glycophosphoprotein that belongs to the small integrin-binding ligand N-linked glycoprotein (SIBLING) family and is also involved in wound healing, neovascularization, and amelioration of vascular calcification [34]. PCNA and OPN could be used to evaluate cell proliferation [35,36]. So, we examined the expression of PCNA and OPN by Western blotting, and the results showed that PCNA and OPN expression was significantly decreased (*p* < 0.01) (Figure 2G). Together, these findings indicate that 24 h hypoxia may inhibit cell cycle progression and cell proliferation in bEnd.3 cells.

### 2.3. Hypoxia Downregulates Mcm2 Expression and Knockdown of Mcm2 Inhibits Cell Cycle Progression and Cell Proliferation

MCM2 plays an important role in the pathological process of cancers, but the effect of MCM2 in microvascular endothelial cells under hypoxia remains unclear. RNA-seq and qRT–PCR showed that *Mcm2* expression was significantly decreased under hypoxia compared with the control conditions (*p* < 0.01) (Figure 3A). The protein expression of MCM2 was also significantly decreased (*p* < 0.01) (Figure 3B). To investigate the role of MCM2 in microvascular endothelial cell proliferation, bEnd.3 cells were transduced with shMcm2 and Mcm2-OE, and the results of EdU assays showed that compared with the negative control, shMcm2 significantly decreased the number of EdU-positive cells (*p* < 0.01), while Mcm2-OE had no effect (Figure 3C). The CCK-8 assay showed that shMcm2 inhibited the growth of cells (Figure 3D). Cyclin A binds and activates CDK2 kinases and plays an important role in the S phase of the cell cycle. Immunoblot analysis showed that the knockdown of Mcm2 significantly decreased the protein expression of PCNA, OPN, CDK2, and cyclin A2 (Figure 3E). Taken together, these findings indicate that MCM2 may be involved in the inhibition of microvascular endothelial cell proliferation under hypoxia.

### 2.4. Mcm2 Overexpression Attenuates Hypoxia-Induced Inhibition of Cell Cycle Progression and Cell Proliferation

To explore whether the overexpression of *Mcm2* could attenuate the inhibition of cell cycle progression and cell proliferation caused by hypoxia, bEnd.3 cells were transduced with the Mcm2-OE construct and then cultured under hypoxia for 24 h. Overexpression of *Mcm2* reversed the decrease in the number of EdU-positive cells that were induced by hypoxia (*p* < 0.01) (Figure 4A). Similar results were obtained in the CCK-8 assay (Figure 4B). The protein expression changes of PCNA, OPN, CDK2, and cyclin A2 were reversed by *Mcm2* overexpression (Figure 4C). Together, these results indicate that overexpression of mcm2 attenuated the inhibition of cell cycle progression and cell proliferation caused by hypoxia.

### 2.5. Hypoxia Upregulates miR-212-3p Expression, and MCM2 Is the Direct Target of miR-212-3p

The whole-transcriptome sequencing assay showed that 112 miRNAs were significantly upregulated, and 108 miRNAs were significantly downregulated (*p* < 0.05) in bEnd.3 cells exposed to hypoxia (Appendix A). A volcano map and a heatmap were generated to visualize the top upregulated and downregulated miRNAs (Figure 5A,B). GO and KEGG enrichment analyses of the total target genes for differentially expressed miRNAs under hypoxic conditions were performed (Figure 5C,D). Then, we predicted the upstream miRNAs of MCM2 through TargetScan and miRanDa. The Venn diagram showed that eight of one-hundred and twelve significantly upregulated miRNAs targeted MCM2 (Figure 5E). Considering the expression levels and fold changes of these miRNAs, we chose miR-212-3p, which was the most significantly upregulated miRNA with high expression levels. RNA-seq and qRT–PCR showed that miR-212-3p expression was significantly increased under hypoxia compared with the control condition (*p* < 0.01) (Figure 5F). The activity of the MCM2 3′-UTR luciferase reporter sharply decreased after treatment with the miR-212-3p mimic by dual luciferase assay (*p* < 0.01), but no significant changes were found in mutated MCM2 3′-UTR after the same treatment, suggesting that MCM2 was a direct target of miR-212-3p (Figure 5G,H).

### 2.6. miR-212-3p Inhibitor Attenuates the Inhibition of Cell Cycle Progression and Cell Proliferation Caused by Hypoxia

In previous studies, miR-212 was reported to serve as an oncogene or tumor suppressor during the development and metastasis of cancer [24]. To explore the role of miR-212-3p in microvascular endothelial cells, we transduced an miR-212-3p mimic or inhibitor into bEnd.3 cells. The results showed that compared with mimic-NC, the miR-212-3p mimic decreased the number of EdU-positive cells (Figure 6A) and the protein expression of MCM2, PCNA, OPN, CDK2, and cyclin A2 (Figure 6C). Similar results were obtained in the CCK-8 assay (Figure 6B). However, the miR-212-3p inhibitor had no effect on cell proliferation.

Then, bEnd.3 cells were transduced with an miR-212-3p inhibitor and cultured under hypoxia for 24 h. The miR-212-3p inhibitor reversed the decrease in the number of EdU-positive cells (*p* < 0.01) (Figure 6D) and the protein expression of PCNA, CDK2, and cyclin A2 that was induced by hypoxia (Figure 6F). Similar results were obtained in the CCK-8 assay (Figure 6E). Together, these results indicated that miR-212-3p inhibited microvascular endothelial cell proliferation, while inhibition of miR-212-3p attenuated the inhibition of cell cycle progression and cell proliferation caused by hypoxia.

### 2.7. miR-212-3p Attenuates Hypoxia-Induced Cell Cycle Progression and Cell Proliferation Inhibition by Regulating MCM2

Since MCM2 was the target of miR-212-3p, we explored whether miR-212-3p could attenuate hypoxia-induced cell cycle progression and cell proliferation inhibition by regulating MCM2. Our results demonstrated that miR-212-3p partially reversed the reduction in the number of EdU-positive cells induced by hypoxia in bEnd.3 cells through MCM2 (*p* < 0.05) (Figure 7A). Western blot assays revealed that inhibition of miR-212-3p could partially promote cell cycle progression and cell proliferation, but this effect was reversed by the knockdown of *Mcm2* under hypoxia (Figure 7B,D). Similar results were obtained in the CCK-8 assay (Figure 7C). In conclusion, these data demonstrated that miR-212-3p attenuated hypoxia-induced cell cycle progression and cell proliferation inhibition by regulating MCM2.

## 3. Discussion

The major finding of this study is that hypoxia inhibits cell cycle progression and cell proliferation in brain microvascular endothelial cells via the miR-212-3p/MCM2 axis. The whole-transcriptome sequencing assay revealed that the cell cycle and DNA replication pathway were significantly downregulated in bEnd.3 cells under hypoxia, and the results were verified by subsequent experiments. The expression of miR-212-3p increased and that of *Mcm2* decreased significantly in bEnd.3 cells under hypoxia. We first found that miR-212-3p could directly regulate the expression of MCM2, which is a critical factor regulating DNA replication and the cell cycle. These results suggest that the miR-212-3p/MCM2 axis may play an important role in maintaining BBB and CNS homeostasis.

Accumulating evidence indicates that BBB will have multiple changes in stroke and high-altitude brain edema, such as degradation of junctional proteins and increased permeability [37,38,39,40]. The endothelium forms the innermost layer of blood vessels and is a multifunctional organ with both systemic and tissue-specific roles [41,42]. Whether hypoxia promotes or inhibits the proliferation of BMECs remains controversial [10,43,44]. Furthermore, the molecular changes in BMECs under hypoxia have not been elucidated. Our study showed that cell proliferation of BMECs was inhibited during hypoxia, and cell cycle, DNA replication, Fanconi anemia, homologous recombination, and mismatch repair pathways were significantly downregulated. Similar to our study, a meta-analysis comprising 430 RNA-seq samples from 43 individual studies including 34 different cell types showed that cell cycle, DNA replication, and DNA repair pathways were repressed by hypoxia [45].

MiR-132 and miR-212 belong to the same family and are located on human chromosome 17 and mouse chromosome 11, respectively. miR-132 and miR-212 are highly conserved among species and have common target genes [19,46]. The whole-transcriptome sequencing assay showed that the expression of miR-132-3p also was increased. In addition, miR-132-3p was predicted to target MCM2. However, despite having the same seed sequence, miR-212 and miR-132 exerted differential effects on endothelial transcriptome regulation and cellular functions with stronger endothelial inhibitory effects caused by miR-212 [47]. In addition, the fold change of miR-212-3p was more obvious. So, we choose miR-212-3p for subsequent experiments. This study showed the expression of miR-212/132 was upregulated in oxygen–glucose deprivation mice and human BMECs as well as in posttraumatic mice and human brain capillaries and serum exosomes [26]. Both miR-132 and miR-212 were significantly increased in the penumbral area 24 h after middle cerebral artery occlusion compared with the sham group [27]. Our results indicated the expression of miR-212-3p increased in BMECs under hypoxia. Another study showed that HIF-1α directly binds to the HRE of the miR-212 promoter by chromatin immunoprecipitation and dual-luciferase assays, and the binding affinity increases under hypoxia [48], which confirms our results.

Previous studies have shown that miR-212 may act as an oncogene or tumor suppressor through different targets or pathways in several cancers, such as hepatocellular carcinoma, pancreatic cancer, and prostate cancer [24,49,50,51]. The miR-212/132 KO mice demonstrated a dramatic increase in retinal vasculature development, indicating that the miR-212/132 cluster had antiangiogenic properties [47,52]. However, it has been reported that blocking miR-212/132 can significantly alleviate the excessive vascular branching phenotype characteristic of zebrafish [52]. The distinct functions performed by miR-212 depend on the targets or pathways involved. The effects of miR-212-3p on the proliferation of BMECs under hypoxia have not been previously reported. Our studies suggested that miR-212-3p could impair the functions of microvascular endothelial cells by targeting MCM2.

MCM2 acts as a vital regulator of DNA replication and the cell cycle by participating in the completion of replication initiation [28,53]. MCM2 was significantly upregulated in almost all cancers and cancer subtypes in The Cancer Genome Atlas and was correlated with the progression and poor prognosis of malignant tumors [31]. MCM2 is thought to be a sensitive biomarker for cancer and a potential novel therapeutic target for cancer treatment [54,55,56]. Here, we reported that the expression of *Mcm2* was decreased under hypoxia and that MCM2 was a direct target molecule of miR-212-3p in vitro. Overexpression of *Mcm2* attenuates the inhibition of cell cycle progression and cell proliferation caused by hypoxia.

In conclusion, miR-212-3p inhibited cell cycle progression and cell proliferation by regulating MCM2 in BMECs under hypoxia. This finding may provide new potential strategies for the treatment of BBB disorder-related cerebrovascular disease.

## 4. Materials and Methods

### 4.1. Cell Culture and Treatment

The mouse brain microvascular endothelial bEnd.3 cell line was obtained from iCell (Shanghai, China). The cells were grown in the specific complete culture medium for bEnd.3 cells (cat. no. iCell-128-0001, iCell, Shanghai, China) containing 10% fetal bovine serum (InCellGene, Burlington, ON, Canada) and 1% penicillin/streptomycin (InCellGene, Burlington, Ontario, NA) at 37 °C in 5% CO_2_. To induce hypoxic injury, bEnd.3 cells were exposed to low oxygen-containing gas (94% N_2_, 5% CO_2_, and 1% O_2_) for 24 h, as previously reported [10], and cells maintained under normoxic conditions were used as the control group. All experiments were repeated 3 times (N = 3).

### 4.2. Whole-Transcriptome Sequencing Assay

Whole-transcriptome sequencing assays were provided by Lianchuan Biotechnology (Hangzhou, China). Total RNA was isolated and purified using TRIzol reagent (Invitrogen, Carlsbad, CA, USA) following the manufacturer’s procedure. The RNA amount and purity of each sample were quantified using a NanoDrop ND-1000 (NanoDrop, Wilmington, DE, USA). Sequencing was performed on an Illumina HiSeq 4000 following the vendor’s recommended protocol. The following procedure for identification of DEGs was used: cutadapt was used to remove the reads that contained adaptor contamination, low-quality bases, and undetermined bases. The sequence quality was verified using FastQC (http://www.bioinformatics.babraham.ac.uk/projects/fastqc/), accessed on 1 May 2022. We used Bowtie2 and Hisat2 to map reads to the genome of species. The mapped reads of each sample were assembled using StringTie. Then, all transcriptomes from samples were merged to reconstruct a comprehensive transcriptome using perl scripts. After the final transcriptome was generated, StringTie and edgeR were used to estimate the expression levels of all transcripts. StringTie was used to perform expression level for mRNAs by calculating FPKM. The differentially expressed mRNAs were selected with log2 (fold change) >1 or log2 (fold change) < −1 and with statistical significance (*p*-value < 0.05) by R package–edgeR. The following procedure for identification of differentially expressed miRNAs was used: Raw reads were subjected to ACGT101-miR (v4.2, LC Sciences, Houston, TX, USA) removed adapter dimers, junk, low complexity, common RNA families, and repeats. Subsequently, unique sequences with length in 18~26 nucleotides were mapped to specific species precursors in miRBase 22.0 by BLAST search to identify known miRNAs and novel 3p- and 5p-derived miRNAs. Differential expression of miRNAs based on normalized deep-sequencing counts was analyzed by selectively using Student’s *t*-test. The significance threshold was set to be 0.01 and 0.05 in each test. To predict the genes targeted by most abundant miRNAs, two computational target prediction algorithms: TargetScan 5.0 (Whitehead Institute for Biomedical Research, Cambridge, MA, USA and Miranda 3.3a (Memorial Sloan-Kettering Cancer Center, New York, NY, USA) were used to identify miRNA binding sites. Finally, the data predicted by both algorithms were combined, and the overlaps were calculated. GO analysis was used for gene function annotation. KEGG analysis was used to determine the biological pathway.

### 4.3. Plasmids and Lentivirus

The lentiviruses shMcm2, Mcm2-OE, mimic-miR-212-3p, inhibitor-miR-212-3p, and negative control were provided by GeneChem (Shanghai, China). The linearization vector was obtained by restriction endonuclease digestion. The target fragment was prepared by primer annealing. The designed primers added restriction sites at both ends. After annealing, the primer and the linear cloning vector contained the same restriction site at both ends. The reaction system was prepared with the linearized carrier and the annealing product, the connection reaction was carried out, and the product was directly transformed. Monoclonals on the plate were selected for PCR identification, and the positive clones were sequenced and analyzed. The correct clone bacterial solution is cultured, and then extracted to obtain a high-purity plasmid for downstream virus packaging.. bEnd.3 cells were transduced with the appropriate lentivirus according to the manufacturer’s instructions. To select stably transduced cells, samples were cultured with puromycin (1 μg/mL) for 3 days or neomycin (3 mg/mL) for 2 weeks. Western blotting and qRT–PCR was performed to determine the levels of *Mcm2* and miR-212-3p.

### 4.4. Quantitative Real-Time PCR (qRT–PCR)

According to the manufacturer’s instructions, total RNA was extracted from bEnd.3 cells with TRIzol reagent (Invitrogen, Carlsbad, CA, USA), and miRNAs were isolated using a SanPrep Column microRNA Extraction Kit (Sangon Biotech, Shanghai, China). The 5× All-In-One RT MasterMix (abm, Richmond, BC, Canada) was used to reverse transcribe mRNA to cDNA. The cDNA and primers were mixed with 2X M5 HiPer SYBR Premix EsTaq (Mei5bio, Beijing, China) and then subjected to qRT–PCR. The levels of miRNAs were confirmed using a miRNA First Strand cDNA Synthesis (Tailing Reaction) Kit and a microRNA qPCR Kit (SYBR Green Method) (Sangon Biotech, Shanghai, China) according to the manufacturer’s instructions. The 2^−ΔΔCt^ method was used to calculate the expression levels of the mRNAs and miRNAs. The following primer sequences were used: 5′-TAACTATGACGGCTCGCTTAAC-3′ (forward) and 5′-ACAGCTACTTTGTTGTCCTTCT-3′ (reverse) for MCM2, 5′-CCGTAAAGACCTCTATGCCAAC-3′ (forward) and 5′- AGGAGCCAGAGCAGTAATCT-3′ (reverse) for Actb, which served as the internal control, and 5′-TAACAGTCTCCAGTCACGGCCA-3′ for mmu-miR-212-3p.

### 4.5. Western Blot Analysis

Cells were lysed in Pierce^TM^ RIPA Buffer (Thermo Fisher Scientific, Waltham, MA, USA) containing a 1% protease inhibitor cocktail (MedChemExpress, Monmouth Junction, NJ, USA) to extract total proteins. The protein concentrations were quantified using a BCA protein assay kit (Thermo Fisher Scientific, Waltham, MA, USA). Equal amounts of protein samples were separated using NuPAGE^TM^ Bis-Tris Gel (Invitrogen, Carlsbad, CA, USA) and transferred to polyvinylidene fluoride (PVDF) membranes. The membranes were blocked with QuickBlock™ Blocking Buffer for Western blotting (Beyotime, Shanghai, China) for 15 min at room temperature followed by incubation at 4 °C overnight with the primary antibodies. The following antibody was used: PCNA (1:1000; CST, Boston, MA, USA), OPN (1:1000; Proteintech, Chicago, IL, USA), CDK1 (1:1000; Proteintech, Chicago, IL, USA), CDK2 (1:5000; Proteintech, Chicago, IL, USA), CDK4 (1:1000; Proteintech, Chicago, IL, USA), cyclin A2 (1:1000; Proteintech, Chicago, IL, USA), cyclin B1 (1:1000; Proteintech, Chicago, IL, USA), cyclin D1 (1:1000; Proteintech, Chicago, IL, USA), cyclin E1 (1:1000; Proteintech, Chicago, IL, USA), MCM2 (1:1000; Proteintech, Chicago, IL, USA), β-actin (1:1000; Proteintech, Chicago, IL, USA). The primary antibodies were diluted with the Primary Antibody Dilution Buffer (Beyotime, Shanghai, China). Then, the membranes were incubated with horseradish peroxidase-conjugated secondary antibodies (InCellGene, Burlington, Ontario, NA) at room temperature for 1.5 h. The secondary antibodies were diluted with the TBST. Finally, the bands were visualized using Western Luminous Substrate (InCellGene, Burlington, Ontario, NA). The blots were analyzed with ImageJ software (v1.53t).

### 4.6. Cell Counting Kit-8 (CCK-8)

Cell viability was evaluated with a CCK-8 assay. bEnd.3 cells were seeded in 96-well plates at a density of 2 × 10^4^ cells/mL (100 µL/well), cultured for 24 h, and then subjected to hypoxia stimulation. Ten microliters of CCK-8 (InCellGene, Burlington, Ontario, NA) was added to each well and cultured for 2 h at 37 °C. The absorbance at 450 nm was examined using a microplate reader.

### 4.7. 5-Ethynyl-2′-deoxyuridine (EdU) Staining

bEnd.3 cells were seeded in 12-well plates and cultured for 24 h. Then, the cells were subjected to hypoxia exposure. Cell proliferation was analyzed with an EdU staining cell proliferation assay kit (RiboBio, Guangzhou, China) according to the manufacturer’s instructions. The nuclei were labeled with Hoechst 33342. The images were observed using an EVOS M5000 Imaging System (Thermo Fisher Scientific, Waltham, MA, USA). The ratio of the number of EdU-positive cells (red) to the total number of Hoechst 33342-positive cells (blue) was used to indicate the percentage of EdU-positive cells. The final value was calculated by normalizing the percentage of EdU-positive cells in five random microscopic fields.

### 4.8. Cell Cycle Analyses

The proportions of cells in the G0/G1, S, and G2/M phases were detected by flow cytometry. bEnd.3 cells were trypsinized and then fixed with 70% cold ethanol at 4 °C for 2 h. Cells were labeled with propidium iodide (PI) according to the manufacturer’s instructions for the Cell Cycle Analysis Kit (Beyotime, Shanghai, China) and detected by flow cytometry.

### 4.9. Luciferase Assay

TargetScan and miRDB were used to predict that the 3′-UTR of MCM2 contains miR-212-3p binding sites. The 293T cells were cotransfected with miR-212-3p reagents (mimic or negative control) and the wild-type (WT) MCM2 3′-UTR or mutant (MUT) MCM2 3′-UTR, which were inserted into the pmirGLO vector using GP-transfect-Mate transfection reagent. After 48 h of cotransfection and incubation, the activity of luciferase was tested using the dual luciferase reporter assay system according to the manufacturer’s instructions, and the activity level of firefly luciferase was standardized to the Renilla luciferase activity coexpressed in each sample.

### 4.10. Statistical Analysis

GraphPad Prism 8.3.0 was used to analyze the data. The data are expressed as the mean ± SEM from at least three independent experiments. The difference between groups was analyzed by an independent t-test, one-way ANOVA, or two-way ANOVA. If the ANOVA showed significant results, post hoc comparisons were conducted. In all cases, *p* < 0.05 and *p* < 0.01 were considered statistically significant.

## Figures and Tables

**Figure 1 ijms-24-02788-f001:**
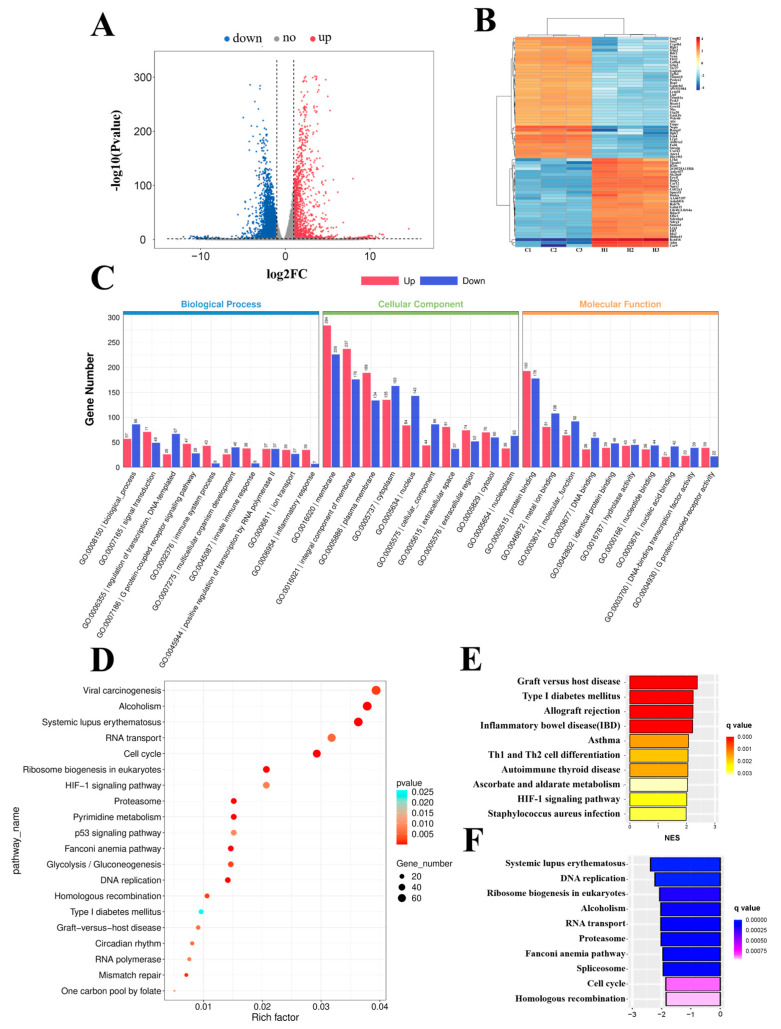
Functional analysis of differentially expressed genes (DEGs) in bEnd.3 cells under 24 h hypoxia. (**A**) Volcano map of differentially expressed genes between the control and hypoxia conditions. (**B**) Heatmap of the top upregulated and downregulated genes. (**C**) Gene ontology (GO) term analysis of differentially expressed genes. (**D**) Top 20 Kyoto Encyclopedia of Genes and Genomes (KEGG) pathways. (**E**) Top 10 upregulated pathways from gene set enrichment analysis (GSEA). (**F**) Top 10 downregulated pathways from GSEA.

**Figure 2 ijms-24-02788-f002:**
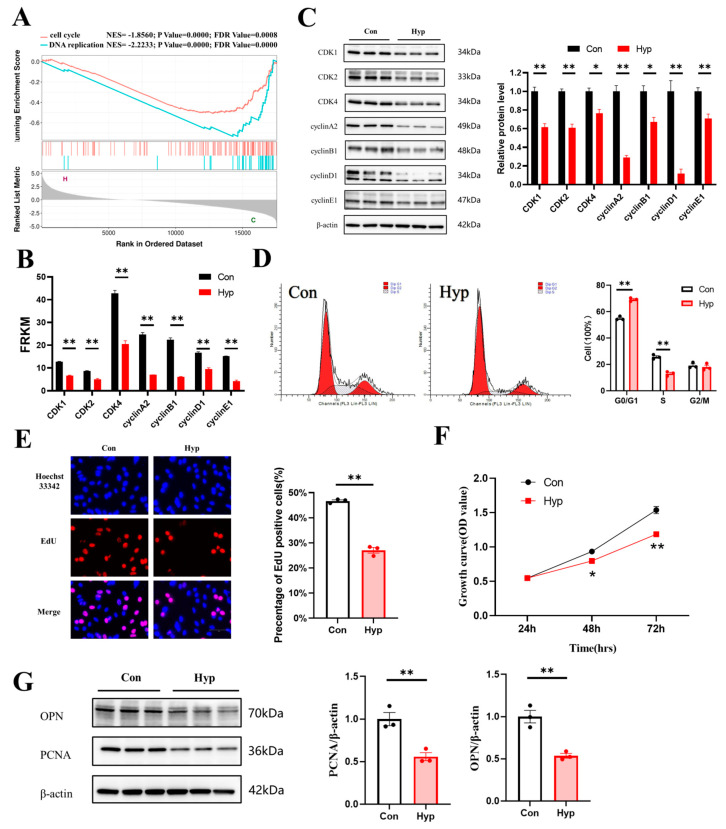
Hypoxia inhibits microvascular endothelial cell cycle progression and cell proliferation. (**A**) GSEA curves of differentially expressed genes for the hypoxia vs. control comparison. (**B**) The expression of cyclin−dependent kinase 1(*Cdk1)*, *Cdk2*, *Cdk4*, *cyclin A2*, *cyclin B1*, *cyclin D1*, and *cyclin E1* from RNA−seq data. (**C**) Western blot analysis of CDK1, CDK2, CDK4, cyclin A2, cyclin B1, cyclin D1, and cyclin E1 (*n* = 3). (**D**) The percentages of cells in the G0/G1, S, and G2/M phases of the cell cycle were quantified by flow cytometry (*n* = 3). (**E**) The EdU (red) assay was used to examine cell proliferation (*n* = 3). EVOS M5000 objective magnification: 40×. (**F**) Cell proliferation was evaluated by a CCK−8 assay (*n* = 3). (**G**) Proliferating cell nuclear antigen (PCNA) and osteopontin (OPN) expression was determined by Western blotting (*n* = 3). Data are shown as the mean ± SEM, * *p* < 0.05, ** *p* < 0.01. Hyp indicates hypoxia, and Con indicates control.

**Figure 3 ijms-24-02788-f003:**
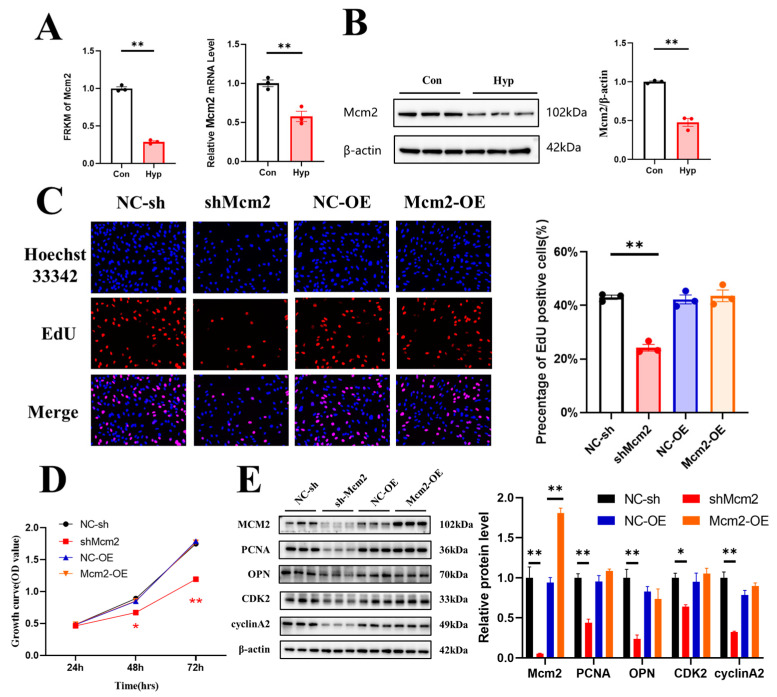
Knockdown of Minichromosome maintenance complex component 2 (Mcm2) inhibits cell cycle progression and cell proliferation. (**A**) The expression of *Mcm2* from RNA-seq data and qRT–PCR (*n* = 3). (**B**) MCM2 expression was determined by Western blotting (*n* = 3). (**C**) The EdU (red) assay was used to examine cell proliferation (*n* = 3). EVOS M5000 objective magnification: 20×. (**D**) Cell proliferation was evaluated by a CCK-8 assay (*n* = 3). The symbol * (red) represents the shMcm2 vs. NC-sh group. (**E**) Western blot analysis of MCM2, PCNA, OPN, CDK2 and cyclin A2 (*n* = 3). Data are shown as the mean ± SEM, * *p* < 0.05, ** *p* < 0.01. NC indicates negative control, and OE indicates overexpression.

**Figure 4 ijms-24-02788-f004:**
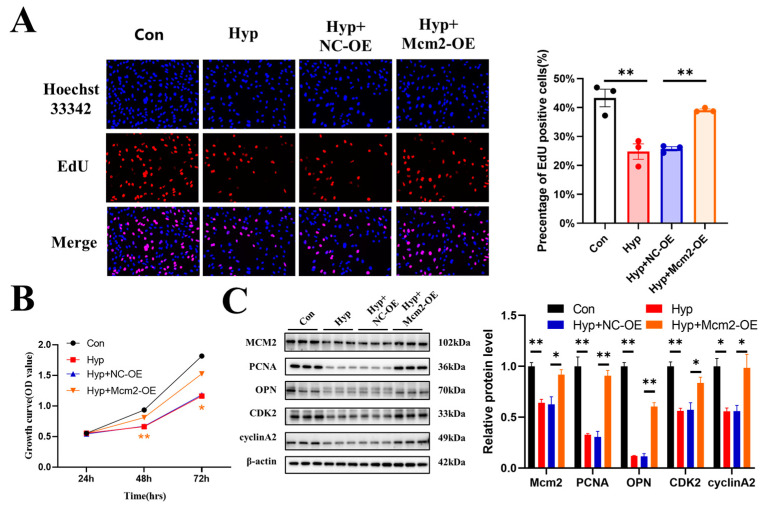
MCM2 attenuates the inhibition of cell cycle progression and cell proliferation caused by hypoxia. (**A**) The EdU (red) assay was used to examine cell proliferation (*n* = 3). EVOS M5000 objective magnification: 20×. (**B**) Cell proliferation was evaluated by a CCK-8 assay (*n* = 3). The symbol * (orange) represents the Hyp + Mcm2-OE vs. Hyp + NC-OE group. (**C**) Western blot analysis of Mcm2, PCNA, OPN, CDK2, and cyclin A2 (*n* = 3). Data are shown as the mean ± SEM, * *p* < 0.05, ** *p* < 0.01.

**Figure 5 ijms-24-02788-f005:**
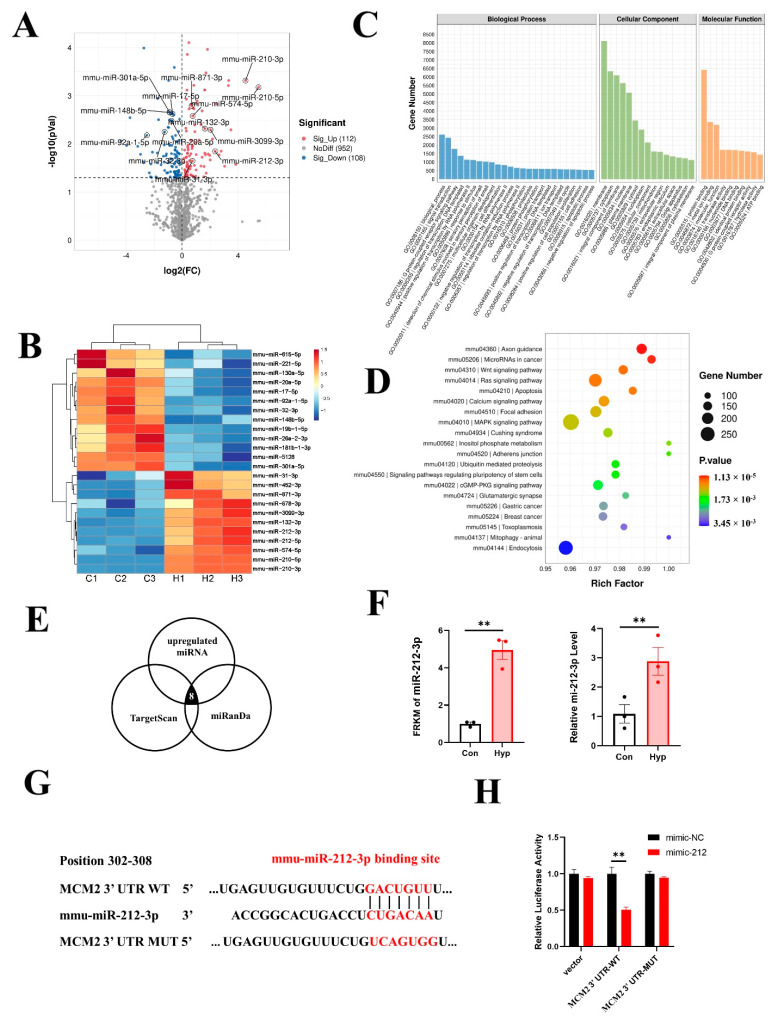
miR-212-3p directly downregulates MCM2 in microvascular endothelial cells under hypoxia. (**A**) Volcano map of differentially expressed miRNAs between the control and hypoxia groups. (**B**) Heatmap of the top upregulated and downregulated miRNAs. (**C**) GO term analysis of target genes for differentially expressed miRNAs. (**D**) KEGG pathways analysis of target genes for differentially expressed miRNAs. (**E**) Venn diagram of significantly upregulated miRNAs targeting MCM2. (**F**) Expression of miR-212-3p from RNA-seq data and qRT–PCR (*n* = 3). (**G**) Schematic diagram of the predicted binding sequences of miR-212-3p to the MCM2 3’UTR WT or MUT. (**H**) Luciferase activity was detected by dual luciferase assay after cotransfection of MCM2 3′UTR-WT/MUT as well as miR-212 mimic and mimic NC in 293T cells. Data are shown as the mean ± SEM, ** *p* < 0.01.

**Figure 6 ijms-24-02788-f006:**
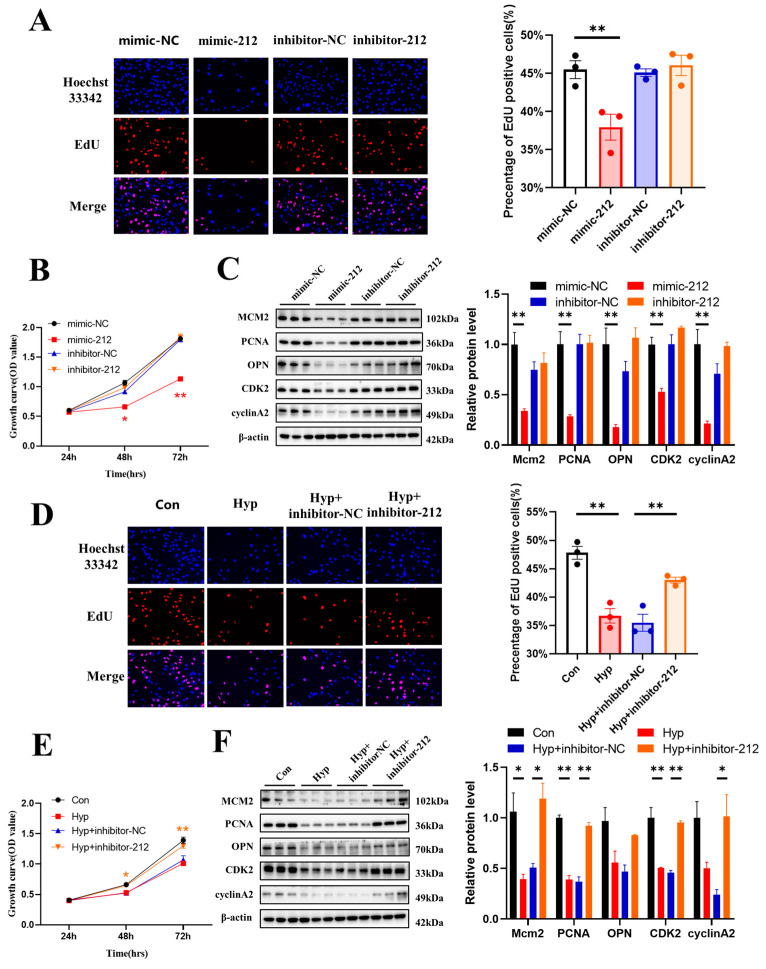
miR-212-3p inhibitor attenuates the inhibition of cell cycle progression and cell proliferation caused by hypoxia. (**A**) The EdU (red) assay was used to examine cell proliferation (*n* = 3). EVOS M5000 objective magnification: 20×. (**B**) Cell proliferation was evaluated by a CCK-8 assay (*n* = 3). The symbol * (red) represents the mimic-212 vs. mimic-NC group. (**C**) Western blot analysis of MCM2, PCNA, OPN, CDK2, and cyclin A2 (*n* = 3). (**D**) The EdU (red) assay was used to examine cell proliferation (*n* = 3). EVOS M5000 objective magnification: 20×. (**E**) Cell proliferation was evaluated by a CCK-8 assay (*n* = 3). The symbol * (orange) represents Hyp + inhibitor-NC vs. Hyp + inhibitor-212. (**F**) Western blot analysis of MCM2, PCNA, OPN, CDK2, and cyclin A2 (*n* = 3). Data are shown as the mean ± SEM, * *p* < 0.05, ** *p* < 0.01.

**Figure 7 ijms-24-02788-f007:**
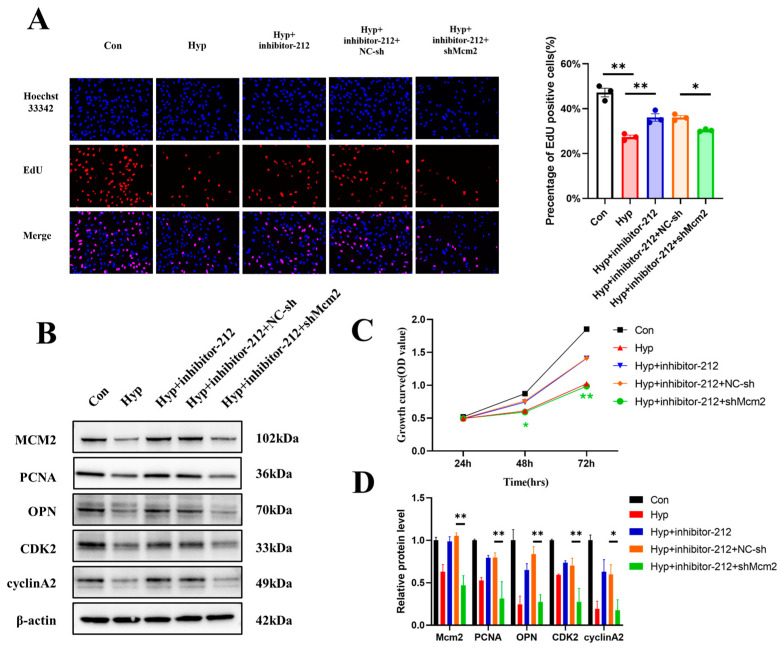
miR-212-3p attenuated hypoxia-induced cell cycle progression and cell proliferation inhibition by regulating MCM2. (**A**) The EdU (red) assay was used to examine cell proliferation (*n* = 3). EVOS M5000 objective magnification: 20×. (**B**,**D**) Western blot analysis of MCM2, PCNA, OPN, CDK2, and cyclin A2 (*n* = 3). (**C**) Cell proliferation was evaluated by a CCK-8 assay (*n* = 3). The symbol * (green) represents Hyp + inhibitor-212 + shMcm2 vs. Hyp + inhibitor-212 + NC-sh. Data are shown as the mean ± SEM, * *p* < 0.05, ** *p* < 0.01.

## Data Availability

Not applicable.

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
