# Peer review of "Hypoxia Inhibits Cell Cycle Progression and Cell Proliferation in Brain Microvascular Endothelial Cells via the miR-212-3p/MCM2 Axis"

_ijms, 2023, doi:10.3390/ijms24032788_

Round 1

Reviewer 1 Report

In this manuscript, the authors found that miR-212-3p inhibited the proliferation of brain microvascular endothelial cells (bEnd.3 cell line) via Mcm2 under hypoxic conditions. Although the experimental results are presented in an easy-to-understand manner, there are several important questions to ask:

Major points;

1. The author performed whole-transcriptome sequencing assay as shown in Figure 1. A number of DEGs were isolated and their functional association determined. I wonder how the authors could select a single gene, namely Mcm2, as a candidate for further experiments.

2. Considering that many genes and various functions are involved in the response to hypoxia, it will be necessary to measure transcriptome changes induced by shMCm2 or miR-212-3p inhibitors under hypoxia.

3. The authors identified 8 genes as targets of miRNAs under hypoxic conditions. I propose that the distribution of total target genes for miRNAs under hypoxic conditions should be presented together with their functional associations.

4. All experiments were performed using bEnd.3 endothelioma. I suggest that hypoxia-induced transcriptome changes should be compared with other cell types to find out that the response to hypoxia is unique to bEnd.3 cells.

5. Procedures for the isolation and functional association of DEGs or miRNAs from high-throughput sequencing data should be described in detail. In addition, the preparation process for shMcm2, Mcm2-OE, mimic-miR-212-3p, inhibitor-miR-212-3p, etc. should be described in detail.

Reviewer 2 Report

In this manuscript Shi et al., focused on the effects of 24h of in-vitro hypoxia (1%O2) on BMEC cell cycle and proliferation. They performed whole-transcriptome sequencing assay and found that hypoxia inhibited cell cycle progression and proliferation and downregulated minichromosome maintenance complex component 2 (Mcm2) expression. Overexpression of Mcm2 attenuated the inhibition by hypoxia. They also found that miR-212-3p was responsible of decreased Mcm2 expression. 

I think the study brings new information about regulation of brain microvascular endothelial cells in in-vitro hypoxia, which needs to be confirmed by in-vivo studies in future. I have some questions and minor concerns about the study:

Did the authors test the hypoxia mediated Mcm2 downregulation by silencing HIF-1a? Does overexpression of Mcm2 has an effect of HIF-1a expression and activity?

Calculations for EdU staining has to be explained.

Abbravations should be used properly.

Statistical significancies should be given in result section at appropriate sections

The reason of investigating the expressions of proliferating cell nuclear antigen

(PCNA) and osteopontin (OPN) by WB should be explained in result section.

Dilutions of primary and secondary antibodies has to be given

HIF-1 expression has to be corrected

Final part of discussion needs to state the importance of the study and has to be conclusive.

Round 2

Reviewer 1 Report

All issues raised in the revised manuscript have been well addressed. Thank you for your efforts.

Reviewer 2 Report

In the revised version of the manuscript authors answered my questions and concerns. I do not have additional suggestions.